# OpenReview forum: "Privacy Awareness for Information-Sharing Assistants: A Case-study on Form-filling with Contextual Integrity"
_TMLR — Accepted by TMLR_

### Review · Reviewer_3MMv · 2025-01-30

**Summary Of Contributions:**

The paper explores operationalizing Contextual Integrity (CI) to steer information-sharing assistants to behave in accordance with privacy expectations. In particular, they use form-filling to guide AI assistants in making privacy-conscious decisions. Experiment results demonstrate the methods works well compared to other baselines (e.g., reasoning) such in protecting privacy.

**Audience:**

Yes

**Claims And Evidence:**

Yes

**Requested Changes:**

Please respond to the Weaknesses mentioned above.

**Strengths And Weaknesses:**

Strengths:
1. The paper is well motivated and clearly written
2. The experiment is comprehensive and well-conducted

Weaknesses:
1. Lack of novelty and technical depth. The proposed method is simply structured output decoding and has been supported by the existing open-sourced framework/API [1, 2]. Technically, it lacks novelty: it just applies existing techniques into a new domain (privacy).
2. Lack of comparison with existing privacy norms. Though the paper discusses privacy in the framework of contextual privacy, other machine learning privacy norms (like differential privacy).
3. Adversarial use cases are not discussed. The framework evaluates privacy basically on its output decoded string. However, privacy should be evaluated under adversarial settings to mimic the worst-case scenarios. For example, how to output logit and probability could cause privacy leakage. In a white-box setting, we should also investigate how the model parameter or intermediate layer representation cause privacy leakage.

[1] https://docs.vllm.ai/en/latest/features/structured_outputs.html
[2] https://platform.openai.com/docs/guides/structured-outputs


Minor:
1 The notations in equations in pages are messy.

---

> ### Author Response · Authors · 2025-02-12
>
> We thank the reviewer for their feedback.
> ### Comparison with existing privacy norms
> We note that most existing machine learning privacy norms such as differential privacy focus on training data leakage (what information from training the model leaks during inference). We consider inference-time leakage (what information provided at inference and never seen during training leaks during training). As such these other notions of privacy do not directly apply to the problem we are considering but we are happy to include any relevant inference-time references that the reviewers could point us to.
> ### Adversarial use cases
> We acknowledge that our designs are susceptible to jailbreaks. The proposed research directions are definitely very promising next steps. They would introduce additional complications and changes in the design of the eval, so we think they would be better suited as contributions in future work.
>
> ### Novelty and contributions
> We thank the reviewer for the useful references. We do not claim anywhere that structured decoding is our contribution. We are building on the advances of structured output decoding to abstract away the problem of privacy reasoning within AI assistants from technical challenges of interactions with complex multi-modal systems. This lets us focus on the actual privacy reasoning capabilities of AI assistants and generalises our work beyond LLM-based systems.
>
> In fact, LLMs as general-purpose agents are mature enough to abstract away most of the technical depths existing in classical ML papers. This allows us to study and steer model behavior on a higher abstraction level, in which we could focus solely on privacy concerns. As such we do not claim algorithmic novelty and can amend any statement that would give this impression. We do believe there is a current need of the community for such evals, and that when assessing this work in comparison to what actually exists at the moment, we think that this is can be a very useful step towards better privacy evals, in particular in the current context where the pace of progress is much faster than that of community-drive and open-source evaluations.
>
> As the other reviewers are noting our contributions include “the application of contextual integrity theory to guide AI assistant behavior [...] a systematic comparison of different architectures for privacy-aware assistants [...] a thorough evaluation framework with multiple metrics and some promising empirical results, demonstrating the benefits of CI-based reasoning” (Reviewer vibd) and “exploring a natural use case for LLM-based personal assistants, that does not have [...] direct representation in prior work (which typically discusses privacy with respect to training data leakage, especially PII in LLM training data, but not data provided explicitly to the model during inference and that it can use in its actions on behalf of the user) [...] a sound methodology for collecting human data as a proxy for widely-accepted norms. The user study design can be useful for future work.” (Reviewer vZ5S)

---

> > ### Comment · Reviewer_3MMv · 2025-02-26
> > **Response to Authors**
> >
> > Thank you for your response. Please incorporate the comparison with other privacy norms and the adversarial use cases in the paper. It will make the paper better.
> >
> > > Re. Novelty and contributions
> >
> >
> > I agree that the paper has its contribution in its scope and real-world applicability. However, my concern is the *novelty and technical depth* of the method rather than the contribution.

---

> > > ### Author Response · Authors · 2025-02-27
> > >
> > > Thank you for the feedback. We have now incorporated the comparison.
> > >
> > > As novelty is not a necessary acceptance criteria  (https://jmlr.org/tmlr/acceptance-criteria.html), we avoid further discussions and thank the reviewer for recognizing the contribution of our work that makes it "of interest" (https://jmlr.org/tmlr/acceptance-criteria.html) to other researchers in this area.

---

### Review · Reviewer_vibd · 2025-02-02

**Summary Of Contributions:**

The paper makes the following contributions:

* Proposes a framework to operationalize contextual integrity (CI) for AI assistants that handle form-filling tasks, aiming to protect user privacy while maintaining utility.
* Introduces different assistant architectures with varying levels of privacy supervision mechanisms
* Develops a synthetic benchmark combining form-filling tasks with human annotations to evaluate privacy-preserving capabilities
* Shows empirical results comparing different assistant approaches and demonstrates that CI-based reasoning achieves better privacy-utility trade-offs

**Audience:**

Yes

**Claims And Evidence:**

Yes

**Requested Changes:**

* Evaluate proposed methods on other external benchmarks.

* Expand the experiments with more diverse models (either open-source or closed-source)

**Strengths And Weaknesses:**

I think this is a timely and interesting paper. First, it introduces the application of contextual integrity theory to guide AI assistant behavior
It also provides a systematic comparison of different architectures for privacy-aware assistants.
Moreover, it proposes a thorough evaluation framework with multiple metrics and some promising empirical results, demonstrating the benefits of CI-based reasoning.

The weaknesses of the paper, majorly, I think lie in its limited Scope and real-world applicability. First, it focuses only on form-filling tasks, which could be too narrow for broad claims about AI assistants. As the concept might work for education and gender, those "explicit" simple information, could fall short in handling complex information (e.g., a layoff plan). Moreover, constructing a form for such complex information unnecessarily induces certain errors which might not be an ideal way to proceed.
I would also want to point out that I didn't see such an agent/assistant framework applied to external benchmarks like ConfAIde, which might cast doubts on the generalizability of the methods.

---

> ### Author Response · Authors · 2025-02-12
>
> We thank the reviewer for their comments. We are really excited about the current development of information sharing assistants and hope that our framework will prove useful for the wide tasks that AI assistants will be used for. We have chosen form-filling as one exemplary application of our framework which we make clear in the abstract of the paper. Would the reviewer suggest we make this focus more explicit in the title?
>
> We would like to explain our decision to focus on form-filling in more detail.
> ### Drawbacks of focusing on too many tasks
> AI assistants are powerful agents that already today automate many tasks on the behalf of the user. As it is not possible to cover all these tasks, we focus on providing one meaningful benchmark that is 1) large enough in size for statistically significant results, 2) limited in scope to make it possible to collect human annotations with limited budget, and 3) designed such that evaluations can be humanly annotated and trusted. Similarly Mireshghallah et al (2023) have focussed on the single mode of meeting notes in their agentic benchmark (Tier 4 of their benchmark).
> ### Form-filling as an excellent first use case of information-sharing assistants
> At the time of writing we believed that form-filling is the most useful and most urgent information sharing task to focus on for AI assistants. Given OpenAI’s recent announcement of their AI assistant, Google’s Project Mariner, and Anthropic’s Claude 3.5 Sonnet, our assumption seems to be proven right, i.e.:
>
> > Project Mariner can understand and reason across everything on your browser screen, including pixels and web elements like text, code, images and **forms**. (https://deepmind.google/technologies/project-mariner/)
>
> > Developers can integrate this API to enable Claude to translate instructions (e.g., “use data from my computer and online to **fill out this form**”) into computer commands. (https://www.anthropic.com/news/3-5-models-and-computer-use)
>
> > Operator can be asked to handle a wide variety of repetitive browser tasks such as **filling out forms**, ordering groceries, and even creating memes. (https://openai.com/index/introducing-operator/)
>
> Forms provide a rich environment for simulating simple information flows of monolithic  facts in many different domains. Note that even the checkout process for ordering groceries can be abstracted away to forms. Meeting note summarisation and action item generation has for example only been considered for a work setting.
> This abstraction allows us to focus on the actual privacy reasoning capabilities of assistants instead of dealing with the assistants’ planning capabilities that would be required for more complex tasks.
> The evaluation of privacy leakage in forms can be really easily assessed with regex. Manual inspection of failure examples has shown that our evaluation does not introduce any noise. Evaluation of more complex tasks such as the meeting note summarisation requires the use of LLM-based evaluation modules that do not necessarily can be trusted.
>
> While forms do not capture all aspects of agent usage, their evaluation allows us to perform a quantitative analysis. We consider form-filling as a necessary first step towards agentic privacy benchmarks, and to the best of our knowledge, this is the first work to propose such an in-depth analysis on the topic and as such it's a significant improvement over the existing literature. This doesn't mean that more work isn't needed to expand the breadth of future analyses.
>
> As the reviewer also noted, our contribution is timely and we thus believe the research community would benefit from the publication of our work.

---

> > ### Comment · Reviewer_vibd · 2025-02-20
> >
> > I think it's valuable to focus on one single task and analyze it in-depth. However, the paper's framing around AI assistants is too broad for investigating form-filling tasks specifically. I would suggest narrowing the scope of the claims.

---

> > > ### Author Response · Authors · 2025-02-21
> > >
> > > Thank you for the feedback. We're happy to shrink the scope both in title and introduction. Specifically, we will add ": A case study on form filling" to the title.

---

### Review · Reviewer_vZ5S · 2025-02-04

**Summary Of Contributions:**

This paper defines and evaluates information-sharing assistants in their ability to provide user information to third-parties as well as to respect the principle of Contextual Integrity (CI). Namely, CI is concerned with limiting information flows by "widely accepted societal norm (e.g. grounded by culture or regulation)". The paper defines information-sharing assistants, state utility and privacy as two objectives, propose a series of LLM-based assistants that attempt to be both useful and privacy preserving (e.g. "self-censoring" just via prompting, via a separate binary judge with and without reasoning, or via a CI-based prompted supervisor), scope a form-filling task / dataset, collect human annotations on the appropriateness of all information flows in the task, and finally evaluate several LLMs (Gemini, Gemma, Mistral models of 2 sizes) on their privacy and utility as form-filling assistants. The authors find a variety of Pareto-optimal assistants with varying trade-offs, with explicit supervision generally being beneficial for privacy.

**Audience:**

Yes

**Claims And Evidence:**

Yes

**Requested Changes:**

[typo] "map the to ground truth" (Page 7)

* I think it's critical to address the validity of the proposed task, since it's fully synthetic and seems to have mostly been produced by manual brainstorming from the authors (listing the forms, fields and personas, with Gemini then used to instantiate the forms and values, paraphrases, etc). This would largely be addressed by using real forms, but perhaps the authors do have more backing for the choices that were made beyond the short description of how they came to be that is in the paper. Otherwise, it's hard to see the value of the experimental evaluation, not knowing how well it captures the real-world problem it's intending to model.

**Strengths And Weaknesses:**

# Strengths

The paper studies an interesting and relevant task, exploring a natural use case for LLM-based personal assistants, that does not have (as far as I'm aware) direct representation in prior work (which typically discusses privacy with respect to training data leakage, especially PII in LLM training data, but not data provided explicitly to the model during inference and that it can use in its actions on behalf of the user).

The authors present a sound methodology for collecting human data as a proxy for widely-accepted norms. The user study design can be useful for future work. It's also interesting to compare model vs human uncertainty, and posit the objective that model refusals should be proportional to how much humans diverge in what to do in a given situation.

# Weaknesses

Although I like the idea of the task, I think the authors need to justify the construction of the benchmark a bit more in order to argue that it contributes to "adding real world complexity into CI evaluation of AI assistant". Since both the forms and personas are synthetically generated, it's rather unclear to me how representative these are of real-world challenges of the task. Some questions this left me wondering:

* What are some real examples of CI violation in the wild? The paper discusses the abstract problem, but it would help the real-world motivation to have at least a couple of real examples in order to show that this is a real task
* Since the paper focuses on widely accepted norms rather than personal preferences, do we know anything about which kind of violation is most common? It seems plausible that most violations occur because of subjective preferences, since (at least legitimate) websites would be discouraged from asking for information that is widely believed to be inappropriate. But perhaps the literature on CI (which I do not know well) has shown how significant this form of violation is.
* About the validity and use of the personas: since the LLM decisions are supposed to only focus on impersonal criteria, are the personas / variation on the data important at all here? Would the task be effectively the same if the LLM had to only produce the DATA_KEY as opposed to the corresponding DATA_VALUE?

---

> ### Author Response · Authors · 2025-02-12
>
> We thank the reviewer for their comments and acknowledging the relevance of our work. To ensure the validity of our benchmark we heavily relied on iterative manual review of the generated data by human annotators of multiple nationalities and diverse backgrounds to vet model outputs; more details below.
> ### Construction of forms
> We ensured the realism of the forms by collecting data keys from real forms, asking human annotators to rate all data keys for their relevance to the form subjects (which were also collected from real forms), followed by manually vetting all generated forms. We iterated over the dataset a number of times. As the reviewer correctly noted, legitimate websites are typically discouraged from asking for inappropriate information. Real forms are also either overly simplistic or complicated in a way that distracts from assessing the privacy-reasoning capabilities of agents. As such the synthetic data approach lets us create forms *at scale* that seem reasonable to human annotators while still ensuring the existence of inappropriate data keys in a controlled way to simulate potentially illegitimate websites. We promise to open-source our forms to convince readers from their realism and back up our claims.
>
> Inappropriate data keys (which we define as unnecessary information) include examples such as gender on job applications, and birthday on newsletters. We prompt the assistants to only fill in fields necessary to achieve the task of the form and as such consider filling in those examples as real-world CI privacy violations. Note that CI is extremely important in such applications – an agent who shares passport number, account password or banking history when making a restaurant booking clearly violates expectations of the user and undermines the purpose of the system.
> ### Synthetic generation of personas
> We opted to use synthetic data to  protect the privacy of users filling in the forms and enable us to share them for reproducibility.  The personas we generated were designed to capture a diverse set of personal backgrounds as outlined in the appendix. Human annotators vetted the personas multiple times to ensure their realism. Apart from that, we have observed that the variance of the models across personas is negligible (Figure 11). As the reviewer correctly noted, we could thus also evaluate the models with placeholder data values, and synthetic personal attributes are thus not a restriction.
>
> We fully acknowledge that we are not using real-world data, and propose to amend the said quote to `We present a novel methodology for evaluating conversational AI assistants through a detailed analysis of form-filling tasks. To achieve this, we employ a range of synthetic personas and use-cases, enabling a rigorous assessment of context-dependent human-AI interaction.` so that we no longer claim real-world complexity.

---

> > ### Comment · Reviewer_vZ5S · 2025-03-01
> >
> > Thank you for the response.
> >
> > OK, I think I'm a bit more convinced of the validity of the setup. It makes sense that it would be hard to find a large number of real examples of these violations in the wild, though I don't doubt that they do exist. In terms of frequency, the vast majority of online forms might not violate CI, but the few cases that do still deserve attention. So, clearly these violations will be over-represented in the synthetic dataset in comparison with the real-world, but as in most things related to privacy, we're more concerned about studying the worst cases anyway, not just being "private in expectation". In that sense, I think this is a valid setup, even though it doesn't reflect the distribution of these issues on the Web.
> >
> > Thinking about the personas, although the values ended up not mattering for the task, on a second though that is important to validate (as the authors already did in Figure 11, I'm just reassessing my original thought on this). It could have been the case that models would be more willing to overshare information from specific demographics, so it makes sense to include the persona variations empirically even if they don't matter abstractly speaking.
> >
> > Assuming the authors clarify these design decisions in the paper, along with the rephrasing of the claims as the authors suggested, I reassessed my evaluation for Claims and Evidence.

---

> > > ### Author Response · Authors · 2025-03-01
> > >
> > > Thank you so much for the constructive feedback. We have now included disclaimers about the distribution of our evaluation examples and the other suggested edits.

---

### Comment · Action_Editor_AbtS · 2025-02-20
**please respond to author comments**

Dear reviewers,

Now authors have posted comments, please engage in discussions.

Thanks,

Your action editor

---

### Comment · Action_Editor_AbtS · 2025-03-01

Dear reviewer 3MMv and vibd,

Thanks a lot for discussing with the authors. Please submit your official recommendation when you are ready.

Thanks,

Your Action Editor

---

### Decision · Action_Editor_AbtS · 2025-03-15

**Recommendation:** Accept with minor revision

**Comment:**

Key revisions include:
1. Narrow scope claims: Adjust the title/introduction to emphasize form-filling as a case study.

2. Synthetic data justification: Expand discussion of benchmark design choices (e.g., why synthetic data was necessary, how it captures real-world violations, and limitations).

3. Comparison with privacy norms: Briefly contrast CI with inference-time privacy notions (e.g., differential privacy) in the related work.

4. Adversarial scenarios: Acknowledge susceptibility to jailbreaking as a limitation and future work.

These revisions are feasible and enhance clarity without requiring major rework.

**Audience:**

The work aligns with TMLR’s focus on privacy and AI ethics. The methodology (CI operationalization, benchmark design, and human annotation framework) will interest researchers studying privacy in AI assistants. The paper’s scope, while narrow (form-filling), provides a foundation for broader applications. The audience will find this relevant.

**Claims And Evidence:**

The claims are supported by well-structured experiments using a synthetic benchmark validated through human annotations. The evidence demonstrates that CI-based reasoning improves privacy-utility trade-offs, though the reliance on synthetic data warrants clearer justification. The authors addressed concerns about realism by iteratively vetting forms/personas and linking examples to real-world CI violations (e.g., unnecessary sharing of gender or birthdate). However, explicit discussion of limitations (e.g., synthetic data distribution vs. real-world scenarios) would strengthen the claims. Claims are supported, but revisions are needed to clarify limitations.